

# From research to applications - Examples of operational ensemble post-processing in France using machine learning

Maxime Taillardat[1,2] and Olivier Mestre[1,2]

[1]Météo-France, Toulouse, France
[2]CNRM UMR 3589, Toulouse, France

**Correspondence:** Maxime Taillardat (maxime.taillardat@meteo.fr)

**Abstract.** Statistical post-processing of ensemble forecasts, from simple linear regressions to more sophisticated techniques, is now a well-known procedure in order to correct biased and misdispersed ensemble weather predictions. However, practical applications in National Weather Services is still in its infancy compared to deterministic post-processing. This paper presents two different applications of ensemble post-processing using machine learning at an industrial scale. The first is a station-based

post-processing of surface temperature in a medium resolution ensemble system. The second is a gridded post-processing of hourly rainfall amounts in a high resolution ensemble prediction system. The techniques used rely on quantile regression forests (QRF) and ensemble copula coupling (ECC), chosen for their robustness and simplicity of training whatever the variable subject to calibration.

    Moreover, some variants of classical techniques used such as QRF or ECC have been developed in order to adjust to oper-

ational constraints. A forecast anomaly-based QRF is used for temperature for a better prediction of cold and heat waves. A variant of ECC for hourly rainfall is built, accounting for more realistic longer rainfall accumulations. It is shown that forecast quality as well as forecast value is improved compared to the raw ensemble. At last, comments about model size and computation time is made.

## 1   Introduction

Ensemble Prediction Systems (EPS) are nowadays well established tools that enable estimate the uncertainty of Numerical Weather Prediction (NWP) models. They can provide a fruitful complement of deterministic forecasts. As recalled by numerous authors (see e.g. Hagedorn et al., 2012; Baran and Lerch, 2018), ensemble forecasts tend to be biased and underdispersed for surface variables such as temperature, wind speed or rainfall. In order to settle bias and misdispersion, ensemble forecasts need to be post-processed (Hamill, 2018).

Numerous statistical ensemble post-processing techniques are proposed in the literature and show their benefits in terms of predictive performance. A recent review is available in Vannitsem et al. (2018). However, the deployment of such techniques into operational post-processing suites is still in its infancy compared to deterministic post-processing. A quite recent review of operational post-processing chains in European National Weather Services (NWS) can be found in Gneiting (2014).





According to their computational abilities, their NWP models to correct, their data policy, and their forecast users and targets, most of the NWS data-science teams have investigated the field of ensemble post-processing with different and complementary techniques, see e.g. Schmeits and Kok (2010); Bremnes (2019); Gascón et al. (2019); Van Schaeybroeck and Vannitsem (2015); Dabernig et al. (2017); Hemri et al. (2016); Scheuerer and Hamill (2018). The transition from calibrated distributions to physically coherent ensemble members has also been examined using Ensemble Copula Coupling (ECC) technique and its derivations, explained in Ben Bouallègue et al. (2016), or variants of the Shaake Shuffle, presented in Scheuerer et al. (2017).

Regarding statistical post-processing for temperatures, a recent non-parametric technique such as Quantile Regression Forests (QRF Taillardat et al., 2016) has shown its efficiency both in terms of global performance and value. Indeed, this method is able to generate any type of distribution because assumptions on the variable subject to calibration is not required. Moreover, this technique chooses itself the most useful predictors to perform calibration. Recently, Rasp and Lerch (2018) have used QRF as a one of the benchmark post-processing techniques.

For trickier variables where the choice of a conditional distribution is less obvious, such as rainfall, van Straaten et al. (2018) have applied successfully QRF for 3-h rainfall accumulations. The QRF approach has been diversified recently, for parameter estimation (Schlosser et al., 2019) and for a better consideration of theoretical quantiles (Athey et al., 2019). In the same vein, Taillardat et al. (2019) have shown that the adjunction of a flexible parametric distribution built on the QRF outputs (named QRF EGP TAIL) compares favorably with state-of-the-art techniques and bring an added value for heavy 6-h rainfall amounts.

In this paper, we propose to present two examples of deployment of ensemble post-processing in the French NWS operational forecasting chain in order to provide gridded post-processed fields. Both examples are complementary:

– A station-based calibration using local QRF of surface temperature on western Europe of the ARPEGE global EPS, associated to an interpolation step and a classical application of ECC.

– A grid-based calibration using QRF EGP TAIL of hourly rainfall on France of the high resolution AROME EPS using (calibrated with rain gauges) radar data, with a derivation of the ECC technique developed for our application.

We also expose some derivations of QRF, QRF EGP TAIL, and ECC techniques in order to take account of extremes prediction, neighborhood management and weather variable peculiarities.

This paper is organized as follows: in Section 2 we describe the two EPS subject to post-processing and their operational configurations. We also describe the predictors involved in post-processing procedures. The Section 3 comprises a short explanation of QRF and QRF EGP TAIL techniques but above all their adjustments set up for an operational and robust post-processing. The Section 4 introduces the post-processing "after post-processing" work. Indeed, we present in this Section the ECC technique and a variant for rainfall intensities. For the post-processing of post-processed temperatures, we exhibit the algorithm of interpolation and downscaling of scattered predictive distributions. The Section 5 show the evaluation of post-





processing techniques through both global predictive performance and day-to-day case study. Our conclusions and a discussion is presented in the Section 6.

## 2    Ensemble prediction systems and data

We present here the French global NWP model ARPEGE, for temperature calibration, and the high resolution limited model area NWP model AROME, for the post-processing of hourly rainfall.

### 2.1    For ARPEGE and ARPEGE EPS

The ARPEGE NWP model (Courtier et al., 1991) is in use since 1994. Its 35-member EPS, called PEARP, is in use since 2004, and a complete description is available in Descamps et al. (2015). These global models have been drastically improved

throughout years and their respective grid scale on western Europe is 5km for ARPEGE and 7.5km for PEARP and forecasts are made 4 times per day from 0 to 108h every 3h. The calibration is performed on more than 2000 stations across western Europe, see Figure 1 for the localisation of these stations on our target grid (called EURW1S100). The gridded data is bilinearly interpolated to the observation locations. The data spans 2 years from 2015, september $1^{st}$ to 2017, august $31^{st}$. The variables involved in the calibration algorithm is provided in Table 1. Operational calibration is currently performed for 2 initialisations

only (6 and 18 UTC). Moreover, predictors coming from the deterministic ARPEGE model are available up to the lead time 60h (except total surface irradiation predictors available from 60h to 78h every 6h).

We can assume that this dataset is less abundant than in Taillardat et al. (2016). This is mainly due to the number of stations to handle and the target grid after interpolation, which is the kilometric AROME grid on western Europe (EURW1S100), composed of more than 4 millon grid points. Since the principle of statistical post-processing is to build some statistical

model linking observations and NWP outputs, two strategies may be considered: the first one is to build a gridded observation archive on the target grid, using scattered station data and some spatialization technique, and estimate statistical models for each gridpoint or each group of gridpoints (block-MOS technique, Zamo et al., 2016). But although block-MOS technique is efficient when dealing with deterministic outputs, preliminary tests (not shown here) are inconclusive regarding post-processing of ensembles. Furthermore, estimating a QRF model for each grid point and each lead time is not adapted to an operational

use, since would involve prohibitive size of constants (around 4 Terabytes in this case) to load and store into memory. The alternative strategy is the following: perform calibration on station data and use a quick spatialisation algorithm, very similar in its principle to regression kriging, in order to produce quantiles on the whole grid. Calibrated members computation involve a ECC phase and the same spatialisation algorithm.

### 2.2    For AROME and AROME EPS

The non hydrostatic NWP model AROME (Seity et al., 2011) is in use since 2007 on the limited area of the Figure 1. The 16-member EPS associated, called PEAROME (Bouttier et al., 2016), is in operational use since the end of 2016. The deterministic model operates on $0.01° \times 0.01°$ EURW1S100 grid whereas the PEAROME runs on a $0.025° \times 0.025°$ grid, that to say about


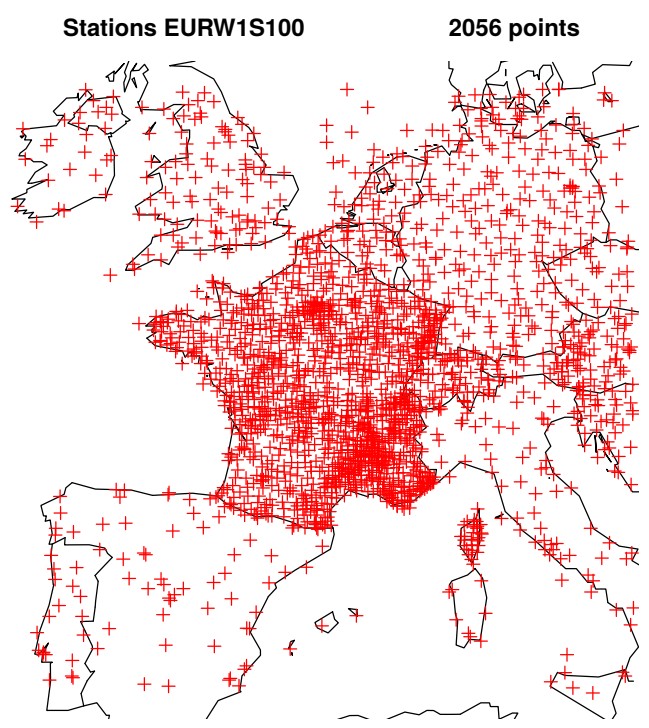

**Figure 1.** Localisation of stations on the target grid.

2.5km. Forecasts are made 4 times a day from 0 to 54h. Data spans 2 years from 2016, december $1^{st}$ to 2018, december
$31^{st}$. The calibration is not performed on the 2.5km grid but on a 10km subgrid. Thus we consider PEAROME here as a
$16 \times 5 \times 5 = 400$-member pseudo-ensemble on a 10km grid. We do this for 3 reasons:

– We solve spatial penalties issues due to the high resolution of the raw EPS (see e.g. Stein and Stoop, 2019).

– We improve ensemble sampling and, we hope, the quality of predictors.

– We reduce computational costs by a factor 25.

The post-processing is realized on these 10km "homogeneity calibration area" (HCA) grid points using the French calibrated
(with rain gauges) radar data ANTILOPE (Laurantin, 2008). Predictors involved in the calibration algorithm are listed in the
Table 2. Note that the temporal penalties due to the high resolution are considered in this choice of predictors. Operational
calibration is currently performed for two initialisations only (9 and 21 UTC) and for lead times up to 45h.

The number of predictors is here less abundant than in Taillardat et al. (2019). This number have been reduced to 25 due to
operational constraints on model size. These predictors has been chosen after a variable selection step using VSURF (Genuer
et al., 2015) and R (Team et al.) package randomForestExplainer (Paluszynska, 2017) among more than 50 predictors. A
complete description of the variable selection is out of scope. To summarize, the most important variables are in average





**Table 1.** Predictors involved in station-based PEARP post-processing

| From ARPEGE model (and up to lead time 60h/78h for irradiation predictors): |
|:---:|
| surface temperature |
| vertical gradient of temperature between surface and 100m |
| surface temperature 3h trend |
| zonal gradient of surface temperature |
| meridian gradient of surface temperature |
| 850hPa potential wet-bulb temperature |
| surface wind speed |
| surface wind direction (factor) |
| sea level pressure |
| mean (on 4 grid point squares) of total cloud cover |
| mean (on 4 grid point squares) of low level cloud cover |
| surface relative humidity |
| accumulated snow depth on ground |
| 3h total surface irradiation in infrared wavelengths |
| 3h total surface irradiation in visible wavelengths |
| **From PEARP (ARPEGE EPS) model:** |
| mean of surface temperature |
| median of surface temperature |
| minimum of surface temperature |
| maximum of surface temperature |
| second decile of surface temperature |
| eighth decile of surface temperature |
| freezing probability |
| **Others:** |
| month of the year (factor) |

minimal and maximal rainfall intensities. These variables are followed by "synoptic" variables such as wind or humidity at medium-level and potential wet-buld temperature. The other variables such as ICA or variables representing the shape of the raw distribution of precipitation are less decisive in average. Variables not retained in the selection precedure are redundant with the main predictors, as other convection indices, medium-level geopotential, and low-level cloud cover, or surface variables. For each HCA, the quantile regression algorithm exhibits 400 quantiles, attributed to each member of each grid point of the HCA after a derivation of ECC technique. The value of grid points and members overlapping two or more HCA are averaged.





**Table 2.** Predictors involved in HCA-based PEAROME post-processing

| From HCA-PEAROME pseudo ensemble: |
| :---: |
| mean of hourly rainfall |
| median of hourly rainfall |
| first decile of hourly rainfall |
| ninth decile of hourly rainfall |
| maximum of hourly rainfall |
| standard deviation of hourly rainfall |
| probability of rain |
| probability of rain $> 5mm.h^{-1}$ |
| maximum of hourly rainfall at previous lead time |
| probability of rain at previous lead time |
| first decile of maximum radar reflectivity |
| ninth decile of maximum radar reflectivity |
| mean of convective available potential energy |
| mean of 850hPa potential wet-bulb temperature |
| first decile of 500m relative humidity |
| ninth decile of 500m relavtive humidity |
| first decile of 700hPa relative humidity |
| ninth decile of 700hPa relavtive humidity |
| first decile of total cloud cover |
| ninth decile of total cloud cover |
| mean of surface wind gust speed |
| mean of 700hPa zonal component of wind speed |
| mean of 700hPa meridian component of wind speed |
| mean of 700hPa wind speed |
| mean of ICA (AROME Convection Index) |

ICA is roughly the product of the modified Jefferson index (Peppier,
1988) with the maximum between 950hPa convergence and maximal
vertical velocity between 400 and 600hPa.

## 3 Calibration techniques

We make here a brief explanation of the QRF and QRF EGP TAIL algorithms before the presentation of their operational
adjustments.





### 3.1 Quantile Regression Forests -based techniques

#### 3.1.1 QRF

Based on the work of Meinshausen (2006), QRF rely on building random forests from binary decision trees, in our case classification and regression trees of Breiman et al. (1984). A tree is iteratively partitionning the training data into two groups.

A split is made according to some thresholds for one of the predictors (or according to some set of factors for qualitative predictors) and chosen such as the sum of the variance of the two subgroups is minimized. This procedure is repeated until a stopping criterion is reached. The final groups (called "leaves") contains training observations such as their predictors values are similar. An example of tree with 4 leaves is provided on the top of the Figure 2.

Binary decision trees are prone to robustless predictions. In random forests, Breiman (2001) solves this issue by averaging

over many trees elaborated from a bootstrap sample of the training dataset. Moreover, each split is determined on a random subset of the predictors.

When a new set of predictors $\boldsymbol{x}$ is available (the blue cross in Figure 2), the conditional Cumulative Distribution Function (CDF) is made by the observations $Y_i$ corresponding the the leaves where the values of $\boldsymbol{x}$ lead in each tree. The predicted CDF thus is

$$\widehat{F}(y|\boldsymbol{x}) = \sum_{i=1}^{n} \omega_i(\boldsymbol{x})\mathbf{1}(\{Y_i \leq y\}) \,, \tag{1}$$

where the weights $\omega_i(\boldsymbol{x})$ are deduced from the presence of $Y_i$ in a final leaf of each tree when one follows the path of $\boldsymbol{x}$.

The interested reader can consult for example Taillardat et al. (2016, 2019); Rasp and Lerch (2018); Whan and Schmeits (2018) for detailed explanations, and comparisons with other techniques in a post-processing context.

#### 3.1.2 QRF EGP TAIL

The reader can notice in the equation 1 that the QRF method cannot predict values outside the range of the training observations. For applications focusing on extreme or rare events it could be a strong limitation if the data depth is small. To circumvent this QRF feature, Taillardat et al. (2019) propose to fit a parametric CDF to the observations in the terminal leaves rather than using the empirical CDF in the equation 1. The parametric CDF chosen for this work is the EGPD3 in Papastathopoulos and Tawn (2013) which is an extension of the Pareto distribution. Naveau et al. (2016) show the ability of this distribution to represent

both low, medium and heavy rainfall and its flexibility. Thus, the QRF EGP TAIL predictive distribution is

$$G(y|\boldsymbol{x}) = P_0 + (1 - P_0)\left[1 - \left(1 + \frac{\xi y}{\sigma}\right)^{-\frac{1}{\xi}}\right]^{\kappa} \,, \tag{2}$$

where $P_0$ is the probability of no rain in the QRF output: $\widehat{F}(y = 0|\boldsymbol{x})$. The parameters $(\kappa, \sigma, \xi)$ in equation 2 are estimated via a robust method-of-moment method.



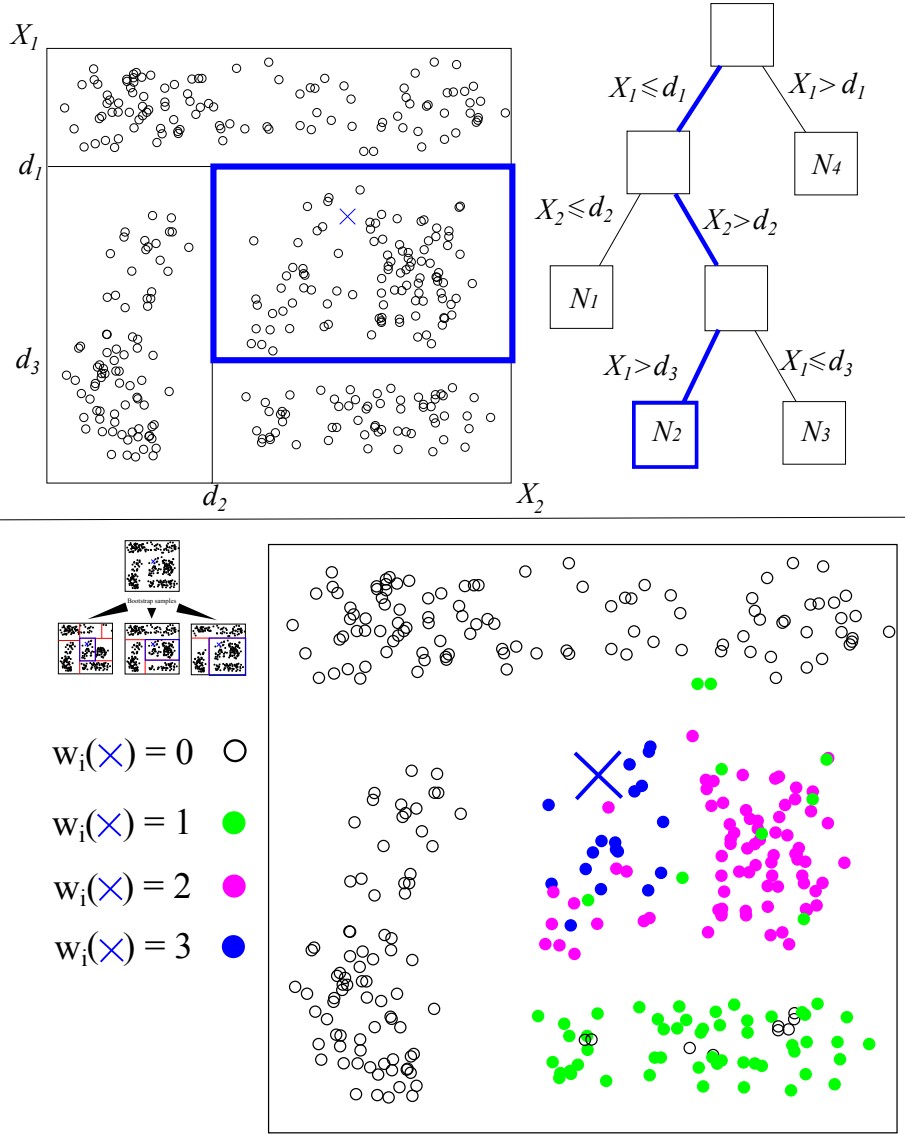

**Figure 2.** Two-dimension example of a (top) binary regression tree and (bottom) three-tree forest. A binary decision tree is built from a bootstrap sample of the data at hand. Successive dichotomies (lines splitting the plane) are made according to a criterion based on observations' homogeneity. For a new set of predictors (the blue cross), the path leading to the corresponding observations is followed. The predicted CDF is the aggregation of the result of each tree





## 3.2 Operational adjustments

### 3.2.1 Temperature

A direct application of QRF algorithm for temperature distribution is suboptimal. Indeed, although QRF is able to return weather-related features such as multi modalities, alternatives scenarios, or skewed distributions, the method cannot go beyond the range of the data. In the operational chain, the QRF algorithm is not trained with observations but with the errors between the observation and the ensemble forecast mean. The result of the equation 1 is in this case the error distribution before translation around raw ensemble mean. The predictive distributions are now constrained by the range of errors made by the ensemble mean. This anomaly-QRF approach generates better distributions than QRF for the prediction of cold and heat waves, and lead (not shown here) to an improvement of about $7\%$ in averaged Continuous Ranked Probability Score (CRPS ; Gneiting and Raftery, 2007), due to this NWP-dependent variable response.

### 3.2.2 Hourly Rainfall

The anomaly-based QRF approach is not employed for hourly rainfall. We think that the choice of a centering variable is as difficult as choosing a good parametric distribution for predictive distributions. In the case of hourly rainfall, the adjustments are not relative to the method but on the construction of the trainig data.

We consider on each HCA predictors calculated with the 400-member pseudo-ensemble. For each HCA, of size $0.1°\times0.1°$, 100 ANTILOPE observations are available. We can consider the observation data as coming from a distribution. Practically speaking, instead of having for each set of predictors one observation $Y_i$, we have in our case $(Y_{i_0}, Y_{i_{25}}, Y_{i_{50}}, Y_{i_{75}}, Y_{i_{100}})$, correspoding of the empirical quantiles of order $0, 0.25, 0.5, 0.75, 1$ of ANTILOPE distribution in the HCA. The length of the training sample is inflated by a factor 5, but it allows to exploit all the information available instead of upscaling high resolution observation data.

## 4 Post-processing... Of post-processing

In this part, we present the techniques employed to transform calibrated predictive distributions into coherent members on grids of interest.

### 4.1 Ensemble Copula Coupling

The Ensemble Copula Coupling method (Schefzik et al., 2013) provide spatiotemporal joint distributions derived from the raw ensemble structure. Its small computational cost makes it for us the privileged way to reorder calibrated marginal distributions, even if other techniques like Schaake Shuffle have their advantages (Clark et al., 2004). Therefore, we make the assumption that on HCA, the structure of the raw ensemble is temporally and spatially sound. Recently, Ben Bouallègue et al. (2016) and Scheuerer and Hamill (2018) propose an improvment of the ECC technique using respectively past data and simulations. In





the context of hourly quantities in hydrology, Bellier et al. (2018) show that perturbations added in the raw ensemble lead to satisfatory multivariate scenarios.

### 4.1.1 ECC for rainfall intensities

As already pinpoint by Scheuerer and Hamill (2018); Bellier et al. (2017), ECC has natural issues with undispersed ensemble and more precisely to attribute precipitation on zero raw members (ie. if the calibrated rain probability $\overline{P_0}$ is greater than the raw one $\overline{F_0}$).

In our case, 400 values have to be attributed in the 16 members of the 25 grid points of the HCA. The procedure, called bootstrapped-constrained ECC (bc-ECC), is as follows:

- If $\overline{F_0} > \overline{P_0}$, a simple ECC is performed.

- If not, we do ECC many times (here 250 times per HCA) and average values.

- Then, a raw zero becomes a non-zero only if there is a raw non-zero in a 3 raw grid point neighborhood.

In this case, $b = 250$ and $c = 3$. The Table 3 gives an example of an HCA of 3 grid points and 2 members.

**Table 3.** Example of bc-ECC ($b = \infty$, $c = 1$) for 2-member (M) ensemble in a 3-grid points (gP) linear HCA.

| In the HCA: | gP1M1 | gP1M2 | gP2M1 | gP2M2 | gP3M1 | gP3M2 |
|---|---|---|---|---|---|---|
| Raw values: | 2 | 0 | 2 | 0 | 5 | 1 |
| HCA Calibrated values: | 0 | 4 | 5 | 5 | 6 | 7 |
| b-ECC and average: | 5.5 | 2 | 5.5 | 2 | 7 | 5 |
| Is rain in M in c gP around ? | - | no | - | yes | - | - |
| Final values: | 5.5 | 0 | 5.5 | 2 | 7 | 5 |

As a result, in a member, post-processing can "dry" grid points, and "wet" grid points if and only if there is a wet grid point close in the raw member. This approach ensures coherent scenarios between post processed rainfall fields and raw cloud cover for example.

## 4.2 Interpolation of scattered post-processed temperature

### 4.2.1 Principle

The problem at hand is challenging:

- The domain covers a large part of western Europe, form coastal regions to Alpine mountainous regions, subject to various climate conditions (oceanic, Mediterranean, continental, Alpine climate).





– Data density is very inhomogeneous (from high density of stations over France, rather dense network over UK, Germany, and Switzerland and sparse density over Spain and Italy).

– Interpolation has to be extremely fast, since more than 1824 high resolution spatial fields have to be produced in a very short time.

Common methods used to interpolate meteorological variables include Inverse Distance Weighting (IDW; Zimmerman et al., 1999), Thin Plate Splines (TPS; Franke, 1982) - both considered as deterministic methods, and kriging (Cressie, 1988), including kriging with external drift to take into account topography effects (Hudson and Wackernagel, 1994). But while IDW suffers
from several shortcomings such as cusps, corners, and flat sports at the data points, preliminary tests showed that both TPS and kriging did not satisfy computation time requirements.

Therefore, a new technique has been developed, very similar to "regression-kriging", whose principle is the following: at station location, perform a regression between post-processed temperatures and raw NWP temperatures using also additional
gridded predictors. The resulting equation is then applied on the whole grid to produce a spatial trend estimation. Regression residuals at station locations are then interpolated. Spatial trend and interpolated residuals are summed to produce the resulting field. Interpolation of residual fields is performed using an automated Multi-level B-splines Analysis (MBA; Lee et al., 1997), an extremely fast and efficient algorithm for interpolation of scattered data.

### 4.2.2    Spatial trend estimation

Several studies have investigated the complex relationships between topography and meteorological parameters, see e. g. Whiteman (2000); Barry (2008). A naive model would be a linear decrease of temperatures with altitude, which is not realistic for temperature at the daily or hourly scale, since vertical profile may be very different than profile of free air temperature. An important phenomenon which was often studied and subject to modelling is cold air pooling in the valleys with diurnal cycle. Frei (2014) uses a change-point model to describe non linear behaviour of temperature profiles.
Topographical parameters include altitude, distance to coast and additional parameters computed following AURELHY method (Bénichou, 1994). AURELHY method is based on a Principal Component Analysis (PCA) of the altitudes. For each point of the target grid, 49 neighboring gridpoint altitudes are selected, forming a vector called landscape. The matrix of landscapes is processed through a PCA. We find out that this method summarizes efficiently topography, since first principal components can easily be interpreted in terms of peak/depression effect (PC1), norther/southern slope (PC2), eastern/western
slopes (PC3) or "saddle effect" (PC4). These AURELHY parameters are presented in the Figure 3.

For interpolation of climate data, most of the time only topographic data is available may play the role of ancillary data to estimate spatial trend. In our case, another important source of information is provided by NWP temperature field at corresponding lead time for each member. As such, PEARP data may not be directly used, since its resolution is coarser than target resolution (0.1°x0.1° rather than 0.01x0.01°). Therefore, PEARP data are projected on the target grid using the following pro-
cedure: for each of the 0.1x0.1 gridpoint, a linear transfer function is estimated through a simple linear regression between each

**Figure 3.** altitude (upper left panel), distance to sea (upper right panel), PC1 to PC4 (from middle left panel to lower right panel).





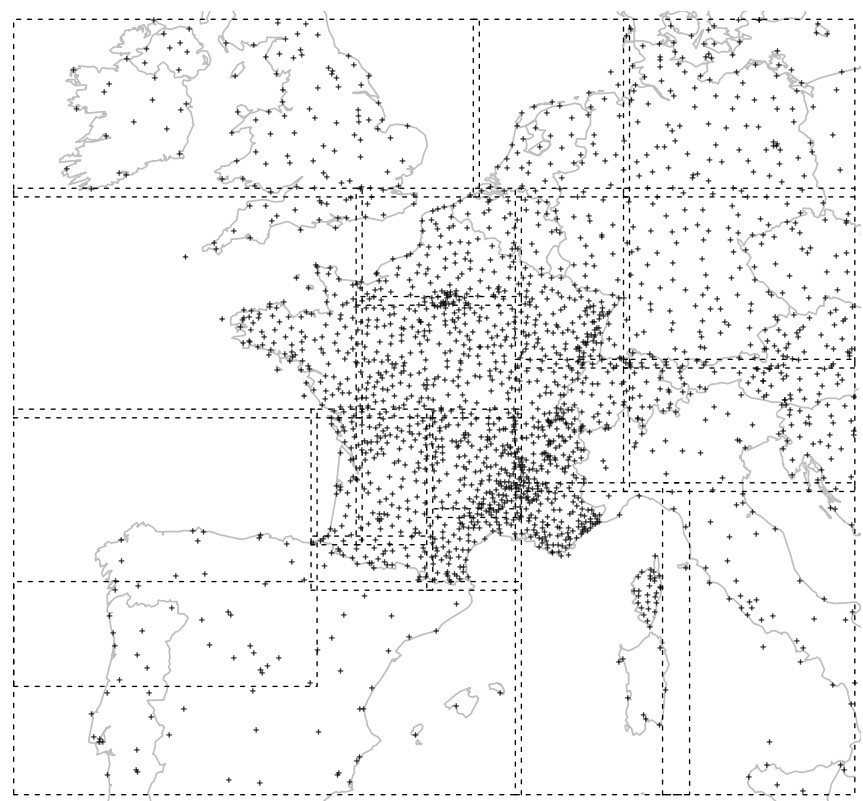

**Figure 4.** Domains used for spatialisation of post-processed temperatures.

of the 100 AROME temperature data (available on the 0.01° resolution grid) and the corresponding ARPEGE data point. Since this relationship is likely to change over seasons and time of the day, those regressions are computed seasonnaly and for every hour of the day, using one year of data. This is a crude but quick way to perform downscaling of PEARP data as shown later.

Since interpolation is to be performed on a very large domain, whose data density vary greatly, several regressions are computed on smaller sub-domains denoted by $D$, whose boundaries are given in Figure 4. Note that size of domains depends on station spatial density. Besides this, domains do overlap: at their intersection, spatial trends are averaged, weights summing up to one and being a linear function of inverse distance to domain frontier. This simple algorithm is very efficient in eliminating any discontinuity between adjacent domains that might appear otherwise.

For a given basetime $b$ and leadtime $t$, validity time is denoted $v$, and season is denoted $S$.

    We denote $\mathrm{alti}_i$, (resp. $\mathrm{d2s}_i$, $\mathrm{PC1}_i, \mathrm{PC2}_i, \mathrm{PC3}_i$, $\mathrm{PC4}_i$) values of altitude (resp. distance to sea, and principal component of elevation 1 to 4) at gridpoint $i$ of the target grid. For every basetime $b$ and leadtime $t$, let $T_k$ be the calibrated temperature





forecast of the kth station point of subdomain $D$, corresponding to gridpoint $i$ of the target grid (0.01°x0.01°) and gridpoint $j$ of PEARP 0.1°x0.1° grid, and $T_j$ the corresponding raw PEARP temperature forecast (same member, basetime and leadtime as $T_k$) at the gridpoint $j$. Then:

$$
\begin{aligned}
T_k &= \beta_{0_D} + \beta_{1_D}(\gamma_{0_{ijvS}} + \gamma_{1_{ijvS}}T_j) & (3) \\
&= \beta_{2_D}\mathrm{alti}_i + \beta_{3_D}(\mathrm{alti}_i - a*_D)\infty\{\mathrm{alti}_i > a*_D\} & (4) \\
&= \beta_{4_D}\mathrm{d2s}_i & (5) \\
&= \alpha_{1_D}\mathrm{PC1}_i + \alpha_{2_D}\mathrm{PC2}_i + \alpha_{3_D}\mathrm{PC3}_i + \alpha_{4_D}\mathrm{PC4}_i & (6) \\
&= \epsilon_k & (7)
\end{aligned}
$$

Term (1) corresponds to the linear influence of the linear projection function of $T_j$ on target gridpoint $i$. Term (2) corresponds to altitude effect, with a possible change in slope of vertical temperature gradient at altitude $a*_D$, whose value is tested on a grid of ten specified elevations for each domain $D$. Third term (3) is the influence of distance to sea. Term (4) is related to first four Principal Components of elevation landscapes. The last term $\epsilon_k$ is the regression residual. Distance to sea predictor does appear only for domains including seashores. Furthermore, domains containing too few station points, namely Spanish and Italian domains have only one predictor, which is linear projection of PEARP temperature data: $\gamma_{0_{ijvS}} + \gamma_{1_{ijvS}}T_j$.

Model estimation of parameters $\beta_{0_D}, \beta_{1_D}, \beta_{2_D}, \beta_{3_D}, \beta_{4_D}, \beta_{1_D}, \alpha_{1_D}, \alpha_{1_D}, \alpha_{1_D}, \alpha_{1_D}$, and $a*_D$ is performed by means of ordinary least squares, model selection being automatically ensured by an AIC procedure.

### 4.2.3 Residuals interpolation

We aim at using an exact, automatic and fast interpolation method for residuals interpolation. Although TPS and kriging may be computed in an automated way, those methods do not meet our criteria in terms of computation time.

Also not an exact interpolation method stricty speaking, the MBA algorithm has been chosen since it is an extremely fast algorithm. Furthermore, the degree of smoothness and exactness of the method may be precisely controlled, as recalled by Saveliev et al. (2005).

A precise description of this method is beyond the scope of this article. We just briefly recall that MBA algorithm relies on a uniform bicubic B-Spline surface passing through the set of scattered data to be interpolated. This surface is defined by a control lattice containing weights related to B-spline basis functions whose sum allows surface approximation. Since there is a tradeoff between smoothness and accuracy of approximation via B-Splines, MBA takes advantage of a multiresolution algorithm. MBA uses a hierarchy of control lattices, from coarser to finer, to estimate a sequence of B-splines approximations whose sum achieves the expected degree of smoothness and accuracy. The reader may refer to Lee et al. (1997) for a complete description of the algorithm.

During tests, we found out that 13 approximations were sufficient to ensure a quasi-exact interpolation (magnitude of errors, around 0.0001°C at station locations), for a visual rendering extremely similar to interpolation TPS, at the cost of a small and



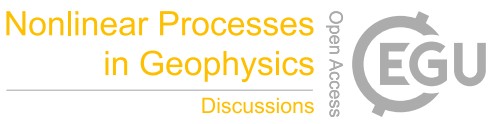

acceptable computing time. Solution with 12 approximations was discarded, since not precise enough (magnitude of errors,

around 0.3°C at station locations): interpolation could not be considered to be exact anymore. When using 14 approximations, computation time dramatically increased.

One important point is that practically, interpolation of residuals is performed once and for all on the whole grid. We found out that undesirable boundary effects could appear at the edges of domains $D$ when residuals were interpolated at each domain $D$ alone.

## 5  General results and day-to-day examples

### 5.1  Temperature

#### 5.1.1  Results of station-wise calibration

We present here the results of the post-processing of PEARP temperature in EURW1S100 stations. For each base and lead time, the Figure 5 shows the averaged CRPS in top panel and PIT statistic mean and $12\times$ variance in bottom panels. This

statistic represents the bias and the dispersion of the rank histograms (Gneiting and Katzfuss, 2014; Taillardat et al., 2016). Subject to probabilistic calibration, the mean of the statistic should be 0.5 and the variance $1/12$, and implies the flatness of rank histograms.

The gain in CRPS is obvious after calibration whatever base and lead times. Moreover, the hierarchy among base times is kept. In both bottom panels, post-processed ensembles are unbiased and well dispersed, contrary to raw ensembles which

exhibits (cold with diurnal cycle) bias and under-dispersion. Nevertheless, we notice that post-processed distributions shows a slight under-dispersion at the end of lead times. This is due to the absence of predictors coming from ARPEGE deterministic model. These predictors do not relate directly with temperature, and thus the addition of weather-related predictors is here crucial for uncertainty accounting. We think that radiation predictors are the most important here, since the presence of these predictor or not is linked to the "roller coaster" behaviour of post-processed PIT dispersion around 3-day lead time.

#### 5.1.2  Performance of interpolation algorithm

Prior any use in spatialisation of post-processed PEARP fields, performances of the interpolation method has been evaluated for deterministic forecasts.

This paragraph is devoted to evaluation of an earlier version of the algorithm over France, that differs only in the fact that NWP temperature field are not available in the predictor set for spatial trend estimation. Benchmarking data consists in 100

forecasts. For each date, 20 cross validation samples are randomly generated, removing 40 points from the full set of points. Original forecast values and interpolated forecast values are then compared, and standard scores (bias, Root Mean Square Error, Mean Absolute Error, 0.95 quantile of absolute error) are computed. Scores are then compared to the COSYPROD interpolation scheme, the previous operational interpolation method. COSYPROD is a quick interpolation scheme, adapted to interpolation at a set of some production points, and derived from IDW method.

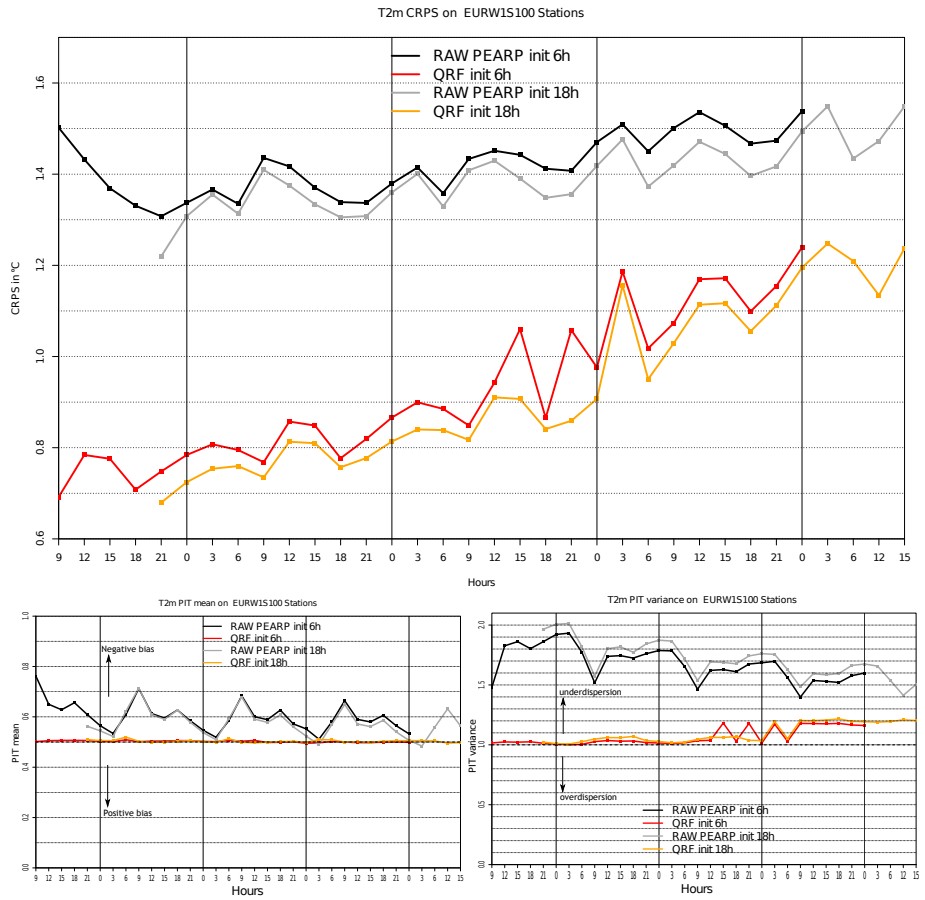

**Figure 5.** Results of PEARP post-processing of temperature in EURW1S100 stations with averages CRPS (top), and mean and variance of PIT statistic, related to rank histograms.

295 Results show that whatever method, bias remains low, but the new spatialisation method outperforms COSYPROD in terms of RMSE, MAE, and .95 quantile of absolute error (Figure 6).

 In addition, the described spatialisation procedure is already used operationnally for interpolation of deterministic temperature forecasts since may 2018. In this application, its performances have been evaluated routinely over a large set climatological station data, that only measure extreme temperatures and do not provide real time data. Hence, this dataset is discarded from

300 any post-processing, but may serve as independant dataset for validation. When comparing forecast performances related to this dataset, increase in root mean square error is around 0.3°C compared to forecast errors estimated at post-processed station data. Hence, this extra 0.3°C root mean square error may be considered as error due to the interpolation process. Note that this is much lower than what has been estimated during the cross-validation phase: all in all, forecast errors and interpolation errors do not add together, but compensate each other to some extent.


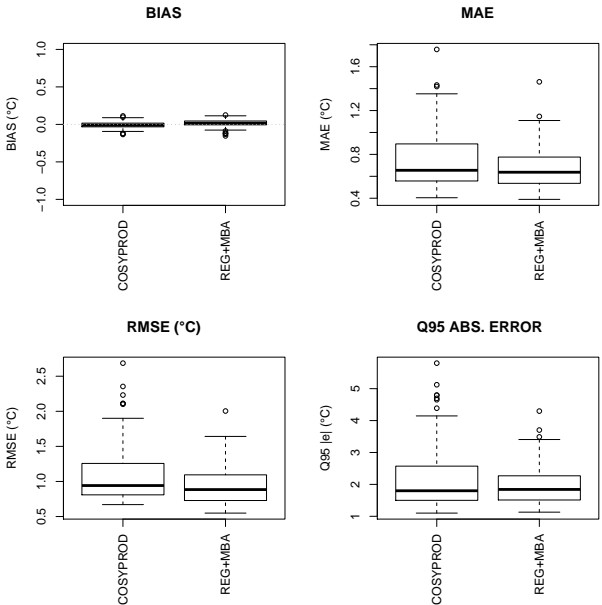

**Figure 6.** Boxplots of Bias (upper left panel), Mean Absolute Error (upper right panel), Root Mean Square Error (lower left panel) and 0.95 quantile for COSYPROD (left boxplot) and new method (right boxplot).

An illustration of the whole procedure is illustrated on PEARP temperatures of basetime 10/03/2019, 18UTC, for lead time 42H.

    Temperature field of raw member 16 is presented in Figure 7, altogether with se same field projected on EURW1S100 grid, according to procedure described in Section 4.2. Estimated spatial trend is shown in Figure 8, and residuals interpolated using MBA procedure with 13 approximation layers can be found in Figure 9.

Resulting field, after calibration, ECC, spatial interpolation phases is presented in Figure 10.

    Same process is repeated here for member 5 (Figure 11).

    One shall note that during the full processing, field values have been modified during the calibration process. But ECC and interpolation are able to keep main features of the original field, that is passage of a front, which is not situated in the same location for both members.

## 5.2   Rainfall

### 5.2.1   Hourly rainfall calibration

Due to high amount of data to handle for evaluation, the scores are presented with averaged lead time and for the base time 9UTC only. In order to make the comparison as fair as possible, the predictive distributions are considered on HCAs and the


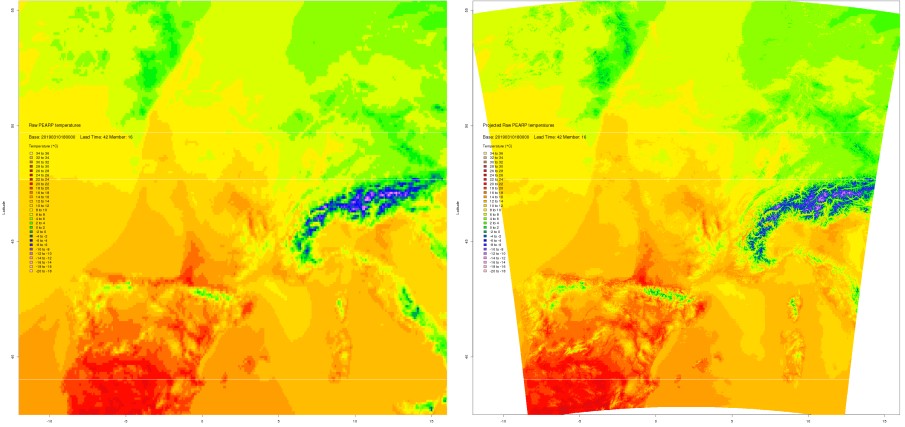

**Figure 7.** Raw member temperatures on 0.1°×0.1° grid (left panel) altogether with raw projected temperatures on 0.01°×0.01° field. Note that this projection can only be done on domain of AROME limited era NWP model.

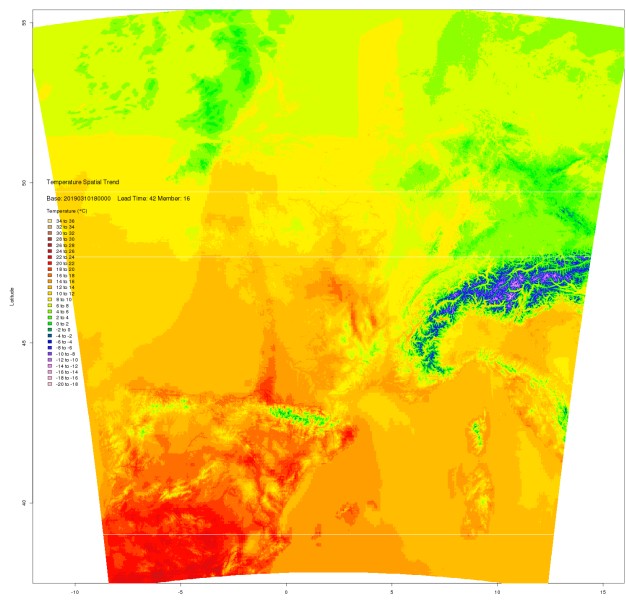

**Figure 8.** Spatial trend estimation using regression model on subdomains as described in Section 4.2.

observation is viewed as a distribution (like in the Section 3.2.2). As a consequence, the divergence of the CRPS should be
used, but the computation of the CRPS on the observations is equivalent (Salazar et al., 2011; Thorarinsdottir et al., 2013).

The Figure 12 shows the averaged CRPS between raw and post-processed ensemble. Here, there is a clear improvment in terms of quality of the forecasts.





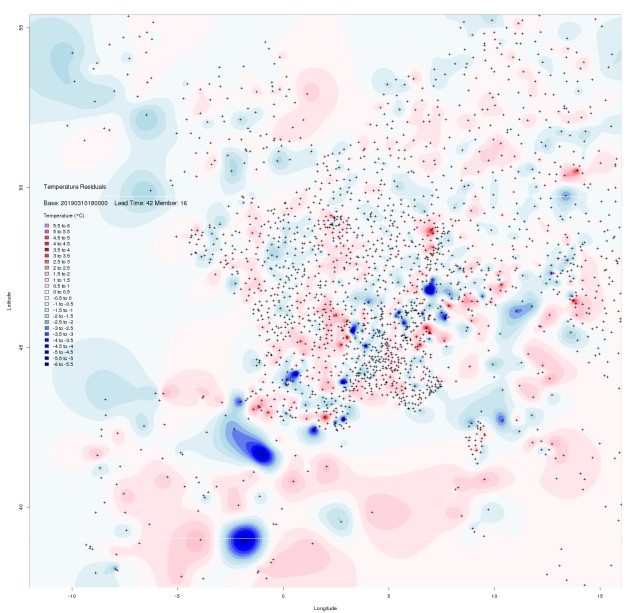

**Figure 9.** Field of residuals interpolated using MBA procedure with 13 layers of approximations.

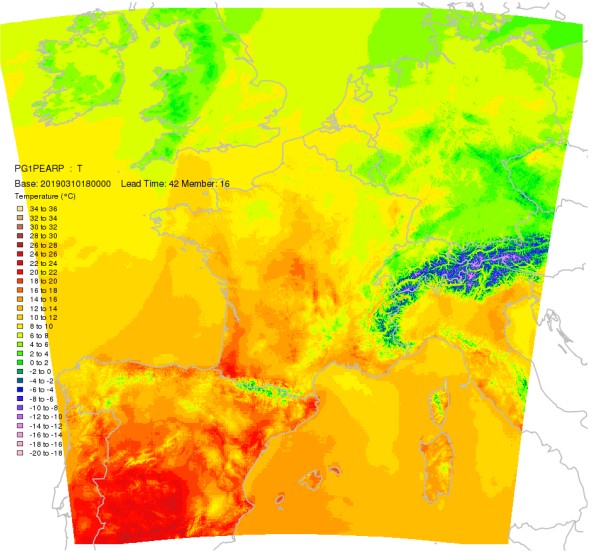

**Figure 10.** Resulting field after the whole procedure.





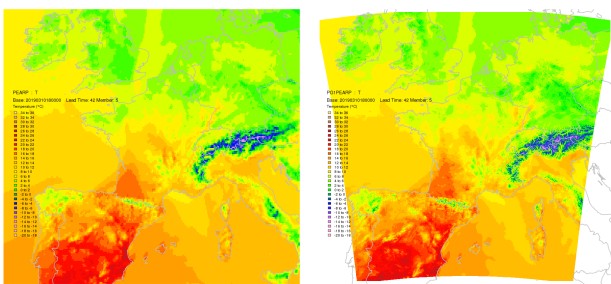

**Figure 11.** Raw PEARP member 6 temperature field (left panel), the same after calibration, ECC and interpolation phase (right panel).

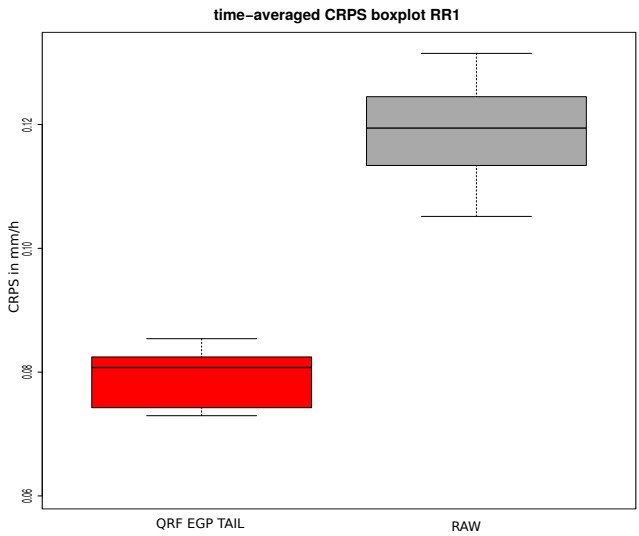

**Figure 12.** Averaged CRPS over lead times of PEAROME hourly rainfall post-processing.

The Figure 13 focuses on the rain event. The top panel shows a ROC curve and a reliability diagram on the same plot. Post-processing improves both resolution and reliability of predictive distributions for the rain event, overpredicted by the raw ensemble. Overprediction of the raw ensemble is also exhibited in the performance diagram (Roebber, 2009) on the bottom panel. Indeed, there is an asymmetry to the top left corner, where frequential bias is more important. And critical success index is increased by 15%, which means that the ratio of rain events (predicted and/or observed) well forecast is improved by 15%.

Concerning higer amounts of precipitation, the focus is put on forecast value. The Figure 14 depicts the maximum of the Peirce Skill Score (PSS ; Manzato, 2007) according to hourly accumulation thresholds. Maximum of the PSS, which corresponds to the nearest point of the top left corner in ROC curves, is a good way to summarize forecast value (Taillardat et al., 2019). We can notice that the improvment is constant after some medium threshold.

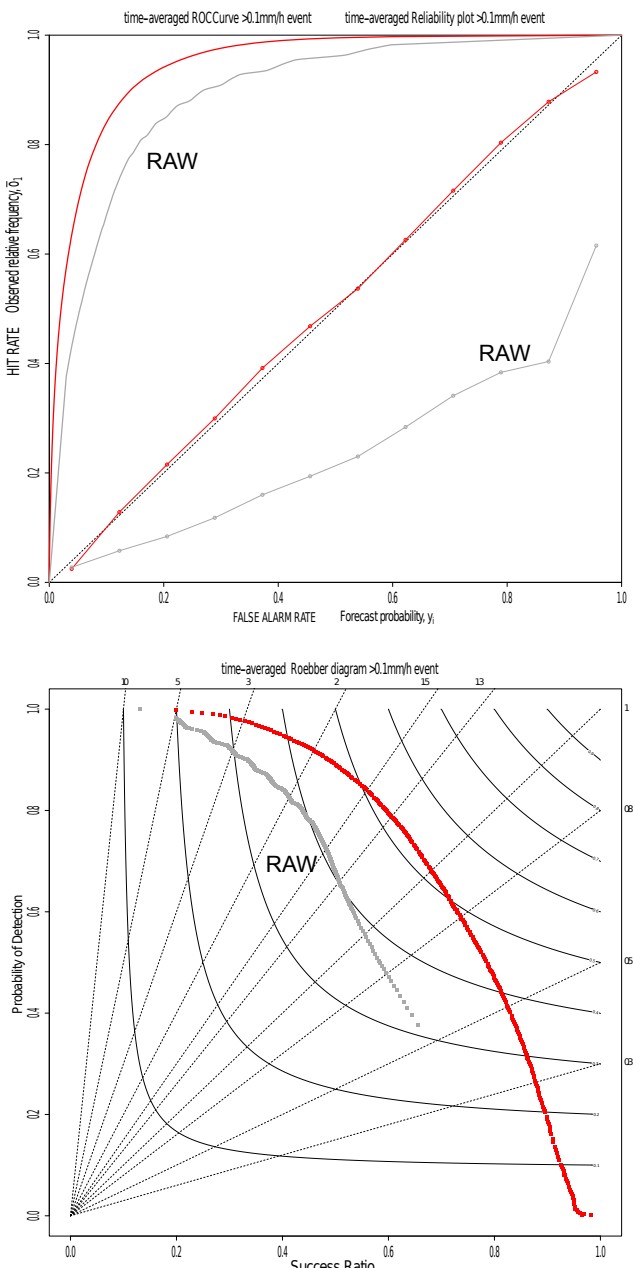

**Figure 13.** ROC curve and reliability diagram (top) and categorical performance diagram (bottom) for the rain event. The raw ensemble suffers from overprediction.



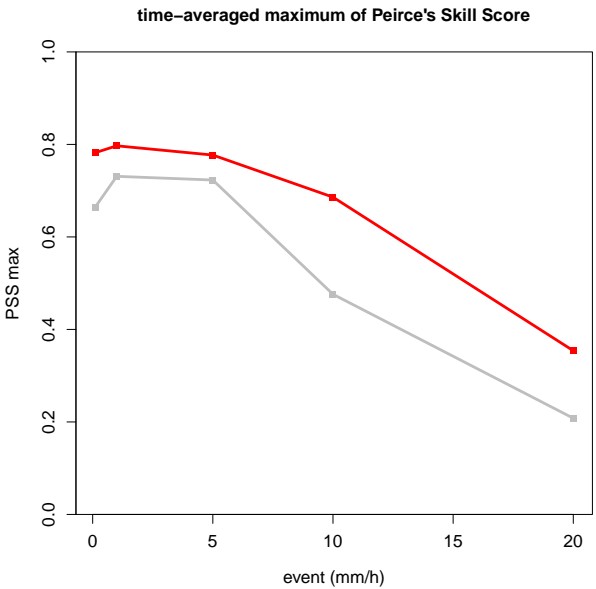

**Figure 14.** Maximum of the Peirce Skill Score among thresholds. The improvment in forecast value tends to be constant.

### 5.2.2 Effects on daily rainfall intensities

We can wonder whether calibrated hourly intensities lead to unrealistic or worse daily rainfall intensities than the raw ensemble. In other words, does the bc-ECC generates cohrent scenarios ? First, we propose in the Figure 15 the comparison of the

predictive quantiles of daily post-processed (after bc-ECC) and raw intensities. The date is 10/22/2019 and realted to a heavy precipitation event in the South of France. Observed accumulations (left of the Figure) reach 300 mm in the day. On the right, the quantiles of order 0.1, 0.5 , and 0.9 of the post-processed ensemble (top right) and raw ensemble (bottm right) are presented. For this event of interest, we see that bc-ECC does not create unrealistic quantities.

A comparison between daily rainfall has been done between raw and post-processed ensembles in the pre-operationnal chain

during October 2019. In the Figure 16, the CRPS of daily distributions shows that bc-ECC does not deteriorate predictive quality. If we divide by 24, we do not obtain the Figure 12 for raw CRPS. Indeed, time penalties disappear with temporal agregation of hourly quantities. The bc-ECC method does not solve temporal agregation. As a consequence, it is not surprising that the daily post-processed CRPS is roughly 24 times the averaged hourly one.

## 6 Conclusion

The two applications described in this article (PEARP temperature and PEAROME rainfall post-processing) are extremely computationally demanding applications, that would fail running on standard workstations in a decent amount of time. Codes



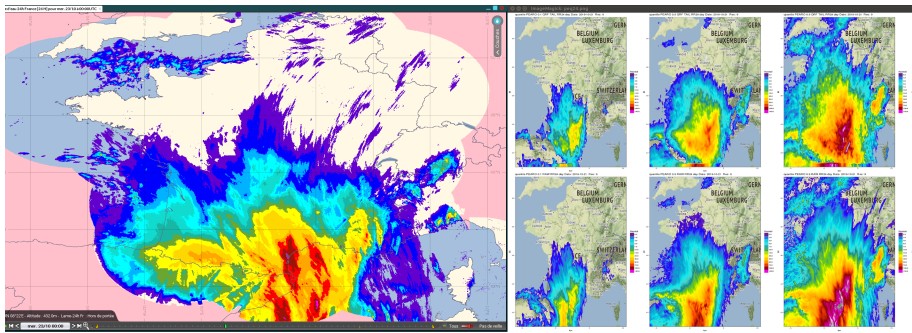

**Figure 15.** Illustration of a heavy precipitation event. On the left, rainfall accumuluted the 10/22/2019, with peaks over 300mm. On the right, quantiles of order 0.1, 0.5, and 0.9 of post-processed (on top) and raw (on bottom) daily rainfall distributions. (Right maps data ©2019 Google)

**Figure 16.** CRPS of daily distributions during October 2019.

are implemented on Météo-France's supercomputer, but even so, a crucial optimization phase has to be achieved, because during the implementation phase, two problems had to be solved:

- The very large number of high resolution fields that has to be produced, since for each lead time a not only statistical
fields (quantiles, mean, standard deviation fields) but also calibrated member fields are computed. This has been achieved using inexpensive but efficient methods such as ECC and MBA, and a massive parallelisation of operations, thanks to R High Performance Computing capabilities. The operational code relies on parallel, foreach, DoSNOW, and DoMC packages, that allow enable OpenMP multicore and MPI multinodes capabilities. The number of cores used in each


node is driven by memory occupation of each process. For example, PEARP temperature uses 4 HPC nodes during 25 minutes, (QRF calibration: 64 cores on 4 nodes (16 cores per node) during 10 minutes, ECC phase: 12 cores on one node during 2 minutes, spatialisation phase: 76 cores on 4 nodes during 15 minutes). PEAROME rainfall uses 162 cores on 18 HPC nodes during 22 minutes for QRF EGP TAIL calibration, and 432 cores on 6 HPC nodes during 3 minutes.

– The huge size of objects produced by quantile regression forests. For a given base time, PEARP temperature application requires to read and load into memory around 300 Gbytes of data, while PEAROME rainfall forests represents more than 600Gbytes of data. Reading this huge amount of data in a reasonable time is possible primarily due to Infiniband network implemented in supercomputer, that features very high throughput and very low latency in I/Os operations. Also, stripping R QRF objects from useless features (regarding prediction) allows substantial save space.

Those two applications now deliver post-processed fields of higher quality than raw NWP fields, and will be used in the future Meteo-France automatic production chain, which is currently in its implementation phase. Post-processed fields are also of higher predictive value, and can lead to great benefits for (trained) human forecasters provided that the dialog between NWP scientists, statisticians and users be strengthened (Fundel et al., 2019).

*Author contributions.* MT developed the station-wise post-processing of PEARP and the post-processing of PEAROME with bc-ECC. OM developed algorithms of interpolation of scattered data and ECC for temperatures. OM configured the operational chain for temperature. OM and MT currently configure the operational chain for rainfall. OM made figures for temperature. MT made figures for rainfall and scores. OM and MT wrote the publication, each rereading the other's part.

*Competing interests.* MT is one of the editors of the Special Issue.

*Acknowledgements.* The authors would thank the team COMPAS/DOP of Météo-France and more particularly Harold Petithomme and Michaël Zamo for their work on R codes. The authors would also thank Denis Ferriol for help during the set up of R codes on the supercomputer.





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
