# Peer review of "From research to applications - Examples of operational ensemble post-processing in France using machine learning"

_Nonlinear Processes in Geophysics, 2019_

## Referee Comment (RC1) · Jonas Bhend (Referee) · 6 Feb 2020

The authors present two applications of machine learning for the operational postprocessing of numerical weather predictions. Their manuscript provides a detailed insight into the full processing chain including state-of-the-art ensemble postprocessing and the many necessary adjustments to fulfill the requirements for high-resolution probabilistic forecasts. Also, the challenges when running a complex ensemble postprocessing in operations are briefly mentioned. The presentation of the results (both text and figures) and their discussion, however, should be reworked for better readability. Therefore, I recommend to accept the manuscript with minor revisions.

[Figure]

**General comments:**

The introduction to the postprocessing methods (Sections 2 - 4) and the constant switching back and forth between temperature and precipitation is difficult to follow. I suggest to either reorganize the discussion to first introduce the complete temperature processing chain and then the rainfall processing chain, or to better guide the reader through the manuscript. For the latter, a more detailed description of the (similarities in) processing across the two cases in the introduction (L 41ff) would be helpful. In particular, this description should reflect the structure of the remainder of the manuscript to manage reader expectations.

The discussion of the operational challenges now in the conclusion should be moved to the discussion section (5). In the conclusion, I instead propose to re-emphasize the benefits and challenges of the postprocessing as being implemented at Meteo France and also try to highlight the aspects of the processing chain that are portable (across parameters, but also to other NHMSs).

The manuscript would clearly benefit from thorough copy-editing. Some of the errors I have mentioned in the section below (e.g. some of the many articles missing), but my edits in that regard are not complete and I therefore encourage the authors to invest into polishing the language. At the very least, please use a spell checker before submission!

**Minor comments:**

L35: as one of the

Section 2.1 and 2.2: why "For", could just be shortened to "ARPEGE and ARPEGE EPS"

L65: throughout the years

L85: on the limited area of Figure 1. The associated 16-member EPS, called PEAROME . . .

L87: for better readability, I suggest to indicate grid resolution only in km as above.

L103: It might be easier to introduce ICA here than in the table

Table1, 2: please indicate the target parameter also in the table caption

L106: exhibits is confusing. Do you mean that 400 samples are drawn corresponding to the 400 ensemble members you have available for ECC? Please rephrase.

L177: Hard to read. Do you mean: The final groups (called "leaves") contain training observations with similar predictor values?

L119: "robustless" doesn't exists nor do I think it applies here. The predictions could be characterized as very robust but are probably prone to overfitting?

Table3: is hard to follow. Could table 3 be made easier by rearranging by member rather than by grid point?

L180: post-processing can introduce rain in a grid box that is dry in raw member only if there is a grid point with rain close by in the raw member.

L216: For the interpolation of climate data, most of the time only topographic data is available which may play. . .

L225: with greatly varying data density

L226: Note that the size

L236ff: the lines should be joined with +

L247: Please specify the impact of model selection. Are generally all predictors used, or is the model usually greatly reduced in complexity due to model selection?
L288: It is not clear to me whether you assess the currently operational spatialization algorithm against COSYPROD (an earlier algorithm) or whether you assess an earlier version of the currently operational spatialization algorithm against COSYPROD (which would predates either of the algorithms).

Figure 5: please redraw this figure for better readability of the labels. The grid lines, which are now quite dominating, could be reduced in weight and the data lines on the other hand should be increased in weight to stand out. Also the legend only has to be shown in the main panel.

Figure 5: Maybe include the number of stations used for evaluation in the figure caption.

L298: large set of climatological

L300: independent

Figure 6: a lot of the figure space is lost to redundant labels. Maybe this could be integrated in one single plot with multiple groups of boxplots?

L305ff: please combine the following sentences into one paragraph

L307: The temperature field . . .. with the same

L316: What is the period for comparison?

L317: Due to the high

L317: I do not understand the rationale to aggregate the scores across lead times. A lead time specific or spatially explicit skill plot would be much more informative.

L320: If the only conclusion drawn from Figure 12 is that postprocessing improves the forecast, Figure 12 could be dropped and the quantitative results could be mentioned in the text (e.g. PP reduced forecast errors by XX-YY

Figure 7-11: I really like the figures and the intent, but I think in the paper, this cannot be presented at the scale one can actually still recognize details. Therefore, I suggest

to rearrange the plots to better illustrate the processing steps and the merits (location of front). To illustrate the postprocessing steps, I suggest to show a zoom-in: one figure with the 5 subpanels (raw member at 0.1, raw member at 0.01, spatial trend, field of residuals, result) for a small area in the French Alps or wherever you deem suited. This could be complemented with a small multiples plot showing the whole domain where the focus is on the position of the front in the raw and postprocessed members.

Figure 12: What is depicted by the box and whiskers? Is it spatial variability or some bootstrap resampling? Please specify. Also, if it is spatial variability, it may be beneficial to show a map (see comment above).

L323: Figure 13 focuses. . .

L323: please specify what exactly constitutes a rain event. Are the same thresholds applied to observations and (postprocessed) forecasts?

L326: frequency bias

L326: Please also discuss the tendency of the postprocessed forecast to underpredict rain events at least in some areas? Is this a feature of the short time period used for evaluation and/or the limited predictability? Is a small or considerable fraction of the forecasts underpredicting rain? Is there an interpretable spatial pattern to it (see comment above on figure 12).

L327: increased by 15

L328: Figure 14 depicts . . .

L331: The conclusion that the improvement in forecast value is constant needs better support. With only two thresholds sampled with similar (absolute) improvement in max PSS, I think this is a bit of a stretch. Also, the absolute improvement is constant, but I assume that for most applications relative improvement would be more relevant.

Figure 13: The size of the axis labels should be increased for better readability. Also,

I would love to see how many cases fall into the bins of the reliability diagram (e.g. the relative contribution via the point size) and in the performance diagram it is not clear where the mass of the distribution is. In case it is many more points than can be visually separated, binning (with bin size corresponding roughly to discernible points in the reliability plot) and point size or some shading with transparency could be used to let the reader focus more strongly on the bulk of the distribution.

Figure 14: Please specify which line is the raw and which the postprocessed forecast.

L333-343: spell check!

L333ff: this section may profit from reversing the order of arguments. First start with the verification of daily rainfall totals (improvement, but relatively less than for hourly rainfall due to the 'disappearance' of timing errors in the raw model forecast). Second, does it also work for extreme events (we don't have the full verification, but present a case study)?

L342: It is not clear to me what is meant by "The bc-ECC method does not solve temporal agregation. As a consequence, it is not surprising that the daily post-processed CRPS is roughly 24 times the averaged hourly one." Is there a problem with temporal aggregation that needs solving? Do you intend to say that because the target is hourly precipitation, the effect of postprocessing is not as beneficial as it could be if daily precipitation were the target of postprocessing?

---

## Referee Comment (RC2) · Anonymous Referee #2 · 14 Feb 2020

The authors have written an interesting paper about the operational implementation of state-of-the-art high-resolution post-processed forecast systems of temperature and precipitation at Meteo France. However, the manuscript can be improved, as outlined in the major and minor remarks below. It also suffers from quite a few grammatical errors and typos. Some are mentioned below, but the authors are advised to carefully reread the manuscript and correct the errors or ask a native speaker to do that before submitting a revised version.

**Major remarks:**

1. For post-processing temperature and precipitation data are used from 2-year periods. However, the paper does not mention which part of the data set is used for training and which part for verification and whether cross-validation has been used or a completely independent verification period is considered as in the verification of daily precipitation amounts (Fig. 16)? Besides, it would also be good to add the verification period in the captions of Figs. 5, 6 and 12-14.

2. The hyperparameters of QRF (like the number of trees and the terminal node size) should be added and whether these are optimized for the training period.

3. I think it is good to add flow diagrams in which all steps involved in the post-processing of temperature and precipitation are displayed.

4. The conclusion section is rather short and there is no discussion paragraph. The current conclusion section mainly focuses on computational aspects, which could be moved to a separate earlier section. Instead it would be good to have a real conclusions and discussion section in which the main conclusions are given and the results are placed in context (in relation to other papers if possible) and in which future work is mentioned.

5. Some of the figures are really small and will become much more clear if they are enlarged, notably Figs. 7-9, 11, 13 and 15. Alternatively the legend in some of the figures can be enlarged to be readable.

**Minor remarks:**

1. Lines 2 and 18: "misdispersed" and "misdispersion" do not seem to be correct English words. Please replace.

2. Line 5: Please add something like "and subsequent interpolation to a grid" after "temperature".

3. Line 40: Please introduce the abbreviation EGP.

4. Lines 44 and 46: Please add references for the two EPS systems.

5. Line 47: Please place "(calibrated with rain gauges)" after "data".

6. Line 53: Please replace "adjustments" by "adjusted".

7. Line 83: Why is it needed to apply the spatialization algorithm twice? Does ECC not account for that?

8. Line 89: Please replace "subgrid" by "grid".

9. Line 91: I would say spatial penalties issues are reduced rather than solved.

10. Lines 94-95: Please replace "calibrated (with rain gauges) radar data ANTILOPE" by "radar data set ANTILOPE (calibrated with rain gauges)" and add the resolution of that data set.

11. Line 100: Please insert "potential" before "predictors".

12. Line 138: Please provide a reference for the method of moment.

13. Line 141: Please insert "forecasting a" before "temperature".

14. Line 159: I would use another title for this section.

15. Line 167: Please replace "data" by "observations".

16. Line 171: The use of "natural" is a bit strange here.

17. Line 181: Please replace "close" by "close by" and add "fields" after "cover".

18. Line 224: Is one year of data enough?

19. Lines 237-240: Please correct the equation: "+" instead of "=" and use the "for all" symbol (?) instead of the infinity symbol.

20. Line 247: Please delete $\beta_{1D}$ the 2nd time is it mentioned and replace the multiple $\alpha_{1D}$'s by $\alpha_{1D}, \alpha_{2D}, \alpha_{3D}, \alpha_{4D}$.

21. Line 248: Please introduce the abbreviation AIC.

22. Line 292-293: Please add a reference for the COSYPROD interpolation scheme.

23. Line 307: It is not necessary to start a new paragraph here.

24. Line 329: Please replace "according to" by "for".

25. Line 331 and caption of Fig. 14: I would not say that the improvement is constant.

26. Axis labels of Fig. 13: Please move 1 of the 2 labels to the other side of the figure (top panel) and add "= 1-FAR" after "Success Ratio" (bottom panel).

27. Caption of Fig. 13: Please add that the red curves are for the post-processed forecasts and add the meaning of the different background curves and dotted lines in the bottom panel (respectively CSI and bias).

28. Line 334: Please replace "propose" by "show".

29. Line 336: Please insert "24-h" after "Observed" and delete "in the day".

30. Line 340: I would say "slightly improves" instead of "does not deteriorate".

31. Line 341: I would replace "Figure 12 for raw CRPS. Indeed," by "same results as in Figure 12 for the raw CRPS, because".

32. Line 342: The bc-ECC method itself does not reduce time penalties because it does not involve temporal aggregation, but I wonder why time aggregation does not have more or less the same effect on the raw and post-processed precipitation forecasts in terms of CRPS?

33. Line 353: Please choose either allow or enable.

34. Line 357: Please add "for bc-ECC" after "3 minutes".

35. Line 362: Please replace "substantial save space" by "to save space substantially".

---

## Author Comment (AC1) · 31 Mar 2020

**Reply to the RC1**

First we would like to thank the RC1, Dr. Jonas Bhend, for his comments and for the time he spent on the paper. You will find our point-to-point response below. It is organized as follows: the initial comments are in blue and the response for each comment in black. For some major comments we answer directly in black after certain sentences. The changes in the manuscript are in red with the line or figure number. If an entire section has been modified or reorganized, the line number is omitted.

General comments:

The introduction to the postprocessing methods (Sections 2 - 4) and the constant switching back and forth between temperature and precipitation is difficult to follow. I suggest to either reorganize the discussion to first introduce the complete temperature processing chain and then the rainfall processing chain, or to better guide the reader through the manuscript. For the latter, a more detailed description of the (similarities in) processing across the two cases in the introduction (L 41ff) would be helpful. In particular, this description should reflect the structure of the remainder of the manuscript to manage reader expectations.

This was a major point of debate between the authors. We have followed your advice and organized Sections 2-4 into one section (Section 2) dealing with the complete temperature processing chain and another section (Section 3) dealing with the complete rainfall processing chain. Moreover, a flowchart for each post-processing procedure has been provided.

The discussion of the operational challenges now in the conclusion should be moved to the discussion section (5). In the conclusion, I instead propose to re-emphasize the benefits and challenges of the postprocessing as being implemented at Meteo France and also try to highlight the aspects of the processing chain that are portable (across parameters, but also to other NHMSs). We agree with this suggestion; the conclusion has been rewritten.

The manuscript would clearly benefit from thorough copy-editing. Some of the errors I have mentioned in the section below (e.g. some of the many articles missing), but my edits in that regard are not complete and I therefore encourage the authors to invest into polishing the language. At the very least, please use a spell checker before submission!

The manuscript will be checked by an English speaker.

Minor comments:

L35: as one of the ok

Section 2.1 and 2.2: why "For", could just be shortened to "ARPEGE and ARPEGE EPS" ok

L65: throughout the years ok

L85: on the limited area of Figure 1. The associated 16-member EPS, called PEAROME . . . ok

L87: for better readability, I suggest to indicate grid resolution only in km as above. ok

L103: It might be easier to introduce ICA here than in the table ok

Table1, 2: please indicate the target parameter also in the table caption ok

L106: exhibits is confusing. Do you mean that 400 samples are drawn corresponding to the 400 ensemble members you have available for ECC? Please rephrase. ok

L117: Hard to read. Do you mean: The final groups (called "leaves") contain training observations with similar predictor values? Yes.

L119: "robustless" doesn't exists nor do I think it applies here. The predictions could be characterized as very robust but are probably prone to overfitting? We have changed this sentence to:
Binary decision trees are prone to unstable predictions insofar as small variations in the learning data can result in the generation of a completely different tree .

Table3: is hard to follow. Could table 3 be made easier by rearranging by member rather than by grid point? ok

L180: post-processing can introduce rain in a grid box that is dry in raw member only if there is a grid point with rain close by in the raw member. ok

L216: For the interpolation of climate data, most of the time only topographic data is available which may play. . . ok

L225: with greatly varying data density ok

L226: Note that the size ok

L236ff: the lines should be joined with + yes

L247: Please specify the impact of model selection. Are generally all predictors used, or is the model usually greatly reduced in complexity due to model selection ? The following sentence has been added. This model selection is influenced by the weather situation, but most often selected variables are the linear projection function of Tj and/or altitude effect – since those two are very well correlated. Distance to sea and PC1 may also be selected quite frequently. PC2 to PC4 selection is much less frequent.

L288: It is not clear to me whether you assess the currently operational spatialization algorithm against COSYPROD (an earlier algorithm) or whether you assess an earlier version of the currently operational spatialization algorithm against COSYPROD (which would predates either of the algorithms).The tested algorithm is the first version of the current operational algorithm – a bit simpler as it does not take into account the linear projection function of Tj. This earlier version is the tester COSYPROD method, which predates our proposed method (both first and current versions). Replaying the full benchmark and adding the linear projection of Tj would have been complex. This benchmark was realized several years ago. Since we added an extra (and valuable) predictor in the full process, we do expect that the conclusions hold for the current operational algorithm – which should even be a bit better than exhibited. We add "current spatialization" to "an earlier version of the algorithm over France" and "which predates both first and current versions".

Figure 5: please redraw this figure for better readability of the labels. The grid lines, which are now quite dominating, could be reduced in weight and the data lines on the other hand should be increased in weight to stand out. Also the legend only has to be shown in the main panel. This figure has been redrawn and labels rescaled.

Figure 5: Maybe include the number of stations used for evaluation in the figure caption. ok

L298: large set of climatological ok

L300: independent ok

Figure 6: a lot of the figure space is lost to redundant labels. Maybe this could be integrated in one single plot with multiple groups of boxplots? There is a new Figure 6 now. Thank you for the suggestion.

L305ff: please combine the following sentences into one paragraph ok

L307: The temperature field . . .. with the same ok

L316: What is the period for comparison? This sentence has been added: the validation is made by a 2-fold cross-validation on the two years of data (one sample per year).

L317: Due to the high ok

L317: I do not understand the rationale to aggregate the scores across lead times. A lead-time-specific or spatially explicit skill plot would be much more informative. This aggregation relies on a huge amount of data to verify and a number of different configurations tested (more than 920). Moreover, different HCA are chosen for each lead time and so inter-lead-time comparisons are difficult. This paragraph is now more detailed.

L320: If the only conclusion drawn from Figure 12 is that postprocessing improves the forecast, Figure 12 could be dropped and the quantitative results could be mentioned in the text (e.g. PP reduced forecast errors by XX-YY. We removed this figure and mention the improvement in the text: The averaged CRPS between the raw and the post-processed ensemble is improved by approximately 30% (from 0.118 to 0.079).

Figure 7-11: I really like the figures and the intent, but I think in the paper, this cannot be presented at the scale one can actually still recognize details. Therefore, I suggest to rearrange the plots to better illustrate the processing steps and the merits (locationof front). To illustrate the postprocessing steps, I suggest to show a zoom-in: one figure with the 5 subpanels (raw member at 0.1, raw member at 0.01, spatial trend, field of residuals, result) for a small area in the French Alps or wherever you deem suited. This could be complemented with a small multiples plot showing the whole domain where the focus is on the position of the front in the raw and postprocessed members. We agree. Figures 7-11 have been replaced by the set of figures suggested. Thank you for this excellent idea.

Figure 12: What is depicted by the box and whiskers? Is it spatial variability or some bootstrap resampling? Please specify. Also, if it is spatial variability, it may be beneficial to show a map (see comment above). Bootstrap resampling. In any case, Figure 12 has been removed.

L323: Figure 13 focuses. . . ok

L323: please specify what exactly constitutes a rain event. It is now defined in the text. Are the same thresholds
applied to observations and (postprocessed) forecasts? Yes.

L326: frequency bias ok

L326: Please also discuss the tendency of the postprocessed forecast to underpredict
rain events at least in some areas? Is this a feature of the short time period used
for evaluation and/or the limited predictability? Is a small or considerable fraction of
the forecasts underpredicting rain? Is there an interpretable spatial pattern to it (see
comment above on figure 12). The categorical performance diagram in Figure 13 is not fully used.
You noticed the tendency of the postprocessed forecast to underpredict rain. We realized that have
not explained the construction of this Figure and that it is confusing. Like the ROC curve, the curve
in the performance diagram is computed for each quantile of the forecast. This latter sentence has
been added. We can assume here that the minimum of predictive distributions nearly never forecast
rain occurrence, but when it does, a false alarm is never made. The minimum of the raw ensemble
detects rain occurrence around 40 times over 100, but when it does this forecast is wrong around 35
times over 100 (1-success ratio). This remark has also been added.

L327: increased by 15 ok

L328: Figure 14 depicts . . . ok

L331: The conclusion that the improvement in forecast value is constant needs better
support. With only two thresholds sampled with similar (absolute) improvement in max
PSS, I think this is a bit of a stretch. This conclusion has been removed. Also, the absolute
improvement is constant, but I assume that for most applications relative improvement would be
more relevant. It is not clear to us what you mean by relative improvement. We guess that it is the
decomposition of the PSS in Hit Rate and False Alarm Rate. If so, a sentence replaced the (too
conclusive) former one by : Most of this improvement is due to the the improvement of the Hit
Rate.

Figure 13: The size of the axis labels should be increased for better readability. Also, I would love
to see how many cases fall into the bins of the reliability diagram (e.g. the relative contribution via
the point size) and in the performance diagram it is not clear where the mass of the distribution is.
In case it is many more points than can be visually separated, binning (with bin size corresponding
roughly to discernible points in the reliability plot) and point size or some shading with
transparency could be used to let the reader focus more strongly on the bulk of the distribution. The
Figure 13 has been redone.

Figure 14: Please specify which line is the raw and which the postprocessed forecast. ok

L333-343: spell check! ok

L333ff: this section may profit from reversing the order of arguments. First start with
the verification of daily rainfall totals (improvement, but relatively less than for hourly
rainfall due to the 'disappearance' of timing errors in the raw model forecast). Second,
does it also work for extreme events (we don't have the full verification, but present a
case study)? This has been done. Thank you for this (logical) suggestion.

L342: It is not clear to me what is meant by "The bc-ECC method does not solve temporal agregation. As a consequence, it is not surprising that the daily post-processed CRPS is roughly 24 times the averaged hourly one." Is there a problem with temporal aggregation that needs solving? Aggregation is not the right word, so we have replaced it with "penalties". Do you intend to say that because the target is hourly precipitation, the effect of postprocessing is not as beneficial as it could be if daily precipitation were the target of postprocessing? Completely. We think that a direct postprocessing of daily precipitation is more effective. It is an intuition that comes from the nature of daily precipitation compared to hourly ones (fewer zeros, smaller variance…). This sentence has been added : Due to the nature of daily, compared to hourly, precipitation distribution (fewer zeros, smaller variance and lighter tail behavior), we believe that a direct post-processing of daily precipitation is more effective if the target variable is daily precipitation.

---

## Author Comment (AC2) · 31 Mar 2020

**Reply to the RC2**

First we would like to thank the RC2 for the time and the work he/she spent on the paper. You will find our point-to-point response to his/her comments below. It is organized as follows: the initial comments are in blue and the response to each comment in black. For some major comments, we answer directly in black after certain sentences. The changes in the manuscript will be in red, with the line number or the figure number. When an entire section has been modified or reorganized, the line number is omitted.

Major remarks:

1. For post-processing temperature and precipitation data are used from 2-year periods. However, the paper does not mention which part of the data set is used for
training and which part for verification and whether cross-validation has been used or
a completely independent verification period is considered as in the verification of daily
precipitation amounts (Fig. 16)? Besides, it would also be good to add the verification
period in the captions of Figs. 5, 6 and 12-14. The validation is made by a 2-fold cross-validation on the two years of data (one sample per year). This sentence has been added in the text and in captions.

2. The hyperparameters of QRF (like the number of trees and the terminal node size)
should be added and whether these are optimized for the training period. This information is now provided at the beginning of the verification section.

3. I think it is good to add flow diagrams in which all steps involved in the post-
processing of temperature and precipitation are displayed. Thank you for this excellent suggestion. Two flowcharts are now available.

4. The conclusion section is rather short and there is no discussion paragraph. The
current conclusion section mainly focuses on computational aspects, which could be
moved to a separate earlier section. Instead it would be good to have a real conclusions
and discussion section in which the main conclusions are given and the results are
placed in context (in relation to other papers if possible) and in which future work is
mentioned. The former conclusion section is now reshaped as the discussion section and a "real" conclusion section has been added.

5. Some of the figures are really small and will become much more clear if they are
enlarged, notably Figs. 7-9, 11, 13 and 15. Alternatively the legend in some of the
figures can be enlarged to be readable. Most of the figures have been redrawn, and labels rescaled. Thank you for this suggestion.

Minor remarks:

1. Lines 2 and 18: "misdispersed" and "misdispersion" do not seem to be correct
English words. Please replace. These terms have been replaced with "poorly dispersed."

2. Line 5: Please add something like "and subsequent interpolation to a grid" after
"temperature". ok
3. Line 40: Please introduce the abbreviation EGP. done

4. Lines 44 and 46: Please add references for the two EPS systems. ok

5. Line 47: Please place "(calibrated with rain gauges)" after "data". ok

6. Line 53: Please replace "adjustments" by "adjusted". This appears to be a misunderstanding of the sentence. The noun form "adjustments" is correct in this case, as it is the subject of the clause.

7. Line 83: Why is it needed to apply the spatialization algorithm twice? Does ECC not account for that? The spatialization algorithm is applied once. A flowchart has been added.

8. Line 89: Please replace "subgrid" by "grid".ok

9. Line 91: I would say spatial penalties issues are reduced rather than solved.

10. Lines 94-95: Please replace "calibrated (with rain gauges) radar data ANTILOPE" by "radar data set ANTILOPE (calibrated with rain gauges)" and add the resolution of that data set. ok

11. Line 100: Please insert "potential" before "predictors". ok

12. Line 138: Please provide a reference for the method of moment. ok

13. Line 141: Please insert "forecasting a" before "temperature". ok

14. Line 159: I would use another title for this section. As the paper has been reorganized, this section has now been omitted.

15. Line 167: Please replace "data" by "observations". ok

16. Line 171: The use of "natural" is a bit strange here. This has been replaced by "innate".

17. Line 181: Please replace "close" by "close by" and add "fields" after "cover". ok

18. Line 224: Is one year of data enough? We are limited by (too frequent) model updates here.

19. Lines 237-240: Please correct the equation: "+" instead of "=" and use the "for all" symbol (?) instead of the infinity symbol. ok

20. Line 247: Please delete β 1D the 2nd time is it mentioned and replace the multiple α 1D 's by α 1D , α 2D , α 3D , α 4D .ok

21. Line 248: Please introduce the abbreviation AIC.ok

22. Line 292-293: Please add a reference for the COSYPROD interpolation scheme. COSYPROD is just the name for an internal IDW-like scheme. IDW references have been provided.

23. Line 307: It is not necessary to start a new paragraph here. ok

24. Line 329: Please replace "according to" by "for". ok

25. Line 331 and caption of Fig. 14: I would not say that the improvement is constant. Also noticed by the RC1. We have removed this statement.

26. Axis labels of Fig. 13: Please move 1 of the 2 labels to the other side of the figure (top panel) and add "= 1-FAR" after "Success Ratio" (bottom panel). Done.

27. Caption of Fig. 13: Please add that the red curves are for the post-processed forecasts and add the meaning of the different background curves and dotted lines in the bottom panel (respectively CSI and bias). Ok, the CSI and bias are now in the caption.

28. Line 334: Please replace "propose" by "show".ok

29. Line 336: Please insert "24-h" after "Observed" and delete "in the day".ok

30. Line 340: I would say "slightly improves" instead of "does not deteriorate". "Slightly improves" would be a bit of a stretch.

31. Line 341: I would replace "Figure 12 for raw CRPS. Indeed," by "same results as in Figure 12 for the raw CRPS, because".ok

32. Line 342: The bc-ECC method itself does not reduce time penalties because it does not involve temporal aggregation, but I wonder why time aggregation does not have more or less the same effect on the raw and post-processed precipitation forecasts in terms of CRPS? For raw daily rainfall amounts, an error in the timing of the shower is made up by the lead time one (or several) hour(s) before or after. Post-processed hourly rainfall amounts are independent (each lead time is post-processed separately). A sentence has been added: Due to

the nature of daily precipitation distribution compared to  hourly ones (fewer zeros, smaller variance and lighter tail behavior), we believe that  direct post-processing of daily precipitation is more effective if the target variable is daily precipitation.

33. Line 353: Please choose either allow or enable. ok

34. Line 357: Please add "for bc-ECC" after "3 minutes".ok

35. Line 362: Please replace "substantial save space" by "to save space substantially".ok